# Psycho-Neuro-Endocrine-Immunology: A Role for Melatonin in This New Paradigm

**DOI:** 10.3390/molecules27154888

**Published:** 2022-07-30

**Authors:** Oscar K. Bitzer-Quintero, Genaro G. Ortiz, Socorro Jaramillo-Bueno, Elsy J. Ramos-González, María G. Márquez-Rosales, Daniela L. C. Delgado-Lara, Erandis D. Torres-Sánchez, Aldo R. Tejeda-Martínez, Javier Ramirez-Jirano

**Affiliations:** 1Centro de Investigación Biomédica de Occidente, Instituto Mexicano del Seguro Social, Guadalajara 44340, Mexico; neuronim26@yahoo.com (O.K.B.-Q.); qfb_lupita@hotmail.com (M.G.M.-R.); atejeda7@hotmail.com (A.R.T.-M.); 2Departamento de Diciplinas Metodológicas y Filosóficas, Centro Universitario de Ciencias de la Salud, Universidad de Guadalajara, Guadalajara 44340, Mexico; genarogabriel@yahoo.com (G.G.O.); daniela.delgadolara@gmail.com (D.L.C.D.-L.); 3Hospital General Regional No 45, Instituto Mexicano del Seguro Social, Guadalajara 44100, Mexico; jsocorro@hotmail.com; 4Unidad de Investigacion Biomedica de Zacatecas, Instituto Mexicano del Seguro Social, Zacatecas 98000, Mexico; elsy.jrg@gmail.com; 5Departamento Académico Formación Universitaria Ciencias de la Salud, Universidad Autónoma de Guadalajara, Guadalajara 44100, Mexico; 6Departamento de Ciencias Medicas y de la Vida, Centro Universitario de la Cienega, Universidad de Guadalajara, Guadalajara 47820, Mexico; erandisdheni@hotmail.com

**Keywords:** immunology, endocrinology, neurology, psychology, melatonin

## Abstract

Psychoneuroendocrinoimmunology is the area of study of the intimate relationship between immune, physical, emotional, and psychological aspects. This new way of studying the human body and its diseases was initiated in the last century’s first decades. However, the molecules that participate in the communication between the immune, endocrine, and neurological systems are still being discovered. This paper aims to describe the development of psychoneuroendocrinoimmunology, its scopes, limitations in actual medicine, and the extent of melatonin within it.

## 1. Introduction

Psychoneuroendocrinoimmunology (PNEI) studies the intimate relationship between immune, physical, emotional, and psychological aspects, states of consciousness, and chemical mediations to demonstrate their psychochemical interdependence. The research of PNEI, initiated in the first decades of the last century, has opened a world of chemical communication between the brain and the endocrine glands that emerged through various scientific figures’ work. Many decades later, peptides released from nerve cells that influence hormone production are still being discovered [1,2].

The hypothalamus is a surprising area of the brain since, in a tiny space, equal to 4 mL in volume, compared to a brain volume of 1350–1500 mL, it contains a set of strategic functions for survival and brain activity organism; a close relationship between the hypothalamus and PNEI has been demonstrated, about its inputs and outputs towards the immune system, its relationship with hormonal processes, with stress states, and with psychological processes. The hypothalamic nuclei regulate water metabolism, hunger and satiety, body temperature, circadian rhythms, emotions and behaviors, and memory. The hypothalamus has one main “gate”, the median eminence, located at the base of the hypothalamus, in the center of the tuber cinereum. The median eminence supplies the pituitary gland with blood rich in hypothalamic hormones and, at the same time, is outside the blood–brain barrier (BBB); it represents an interface with the general blood circulation. Therefore, it functions not only as an exit door but also as a gateway to the hypothalamus for some essential peripheral hormones, such as leptin. The hypothalamus has at its disposal the platform control of the endocrine system, which is achieved through three different axes [3]:(1)By direct secretion, passing through the neurohypophysis, of hormones arginine vasopressin and oxytocin, which affect the kidney, the gravid uterus, the postpartum mammary gland, and the brain [3].(2)Hormones that influence the adenohypophysis to regulate the activity of the gonads, uterus, thyroid, adrenal cortex, liver, and bones [3].(3)Through hormones and neural connections with the autonomic nervous system, which innervates the pineal gland, pancreas, and adrenal medulla [3,4].

All the hypothalamus hormones have a particular objective but are rarely unique because their function is generally not limited only to the endocrine compartment. For example, thyrotropin-releasing hormone (TRH) has its main activity on pituitary endocrine cells that produce thyroid-stimulating hormone (TSH), but TRH also stimulates the release of prolactin, adrenocorticotropic hormone (ACTH), and growth hormone (GH). Thus, somatostatin not only inhibits growth hormone production but also inhibits TSH. Similarly, dopamine inhibits prolactin, TSH, gonadotropins luteinizing hormone (LH), follicle-stimulating hormone (FSH), and occasionally even GH. To complete the polyhedral characteristics of the hypothalamic platform and its extensive influence on the body, the neurons in the paraventricular nucleus and the arcuate nucleus produce hormones and a myriad of neurotransmitters and neuromodulators [3,5,6] (Figure 1).

## 2. Development of Psychoneuroendocrinoimmunology

Research in recent years has revolutionized the medical and psychological view by contributing a profoundly unitary and psychosomatic understanding of the human being and highlighting the intimate relationship between physical, emotional, and psychological aspects. Candace Pert, a researcher at the National Institute of Mental Health (NIMH), discoverer of endorphins and neuropeptides, maintains that it is no longer possible to separate the physical aspect from the mental, but that: “We must speak of mind-body as a single entity integrated” [7,8].

PNEI is inserted as a link between medicine and psychology, with a mind–body model that includes all the physiological, emotional, psychological, behavioral, and social processes understood as an organic and unitary system: a network of systems biology [1].

The origin of the PNEI traditionally dates back to the studies of Walter Bradford Cannon (1871–1945), an American physiologist and psychologist from Harvard University. This took the concept of “milieu Interieur” (i.e., internal organic environment) developed by Claude Bernard (1813–1878), a French physiologist [7]. His research on the relationships between animals studied the emotional implications of stress conditions: fight or flight and the attitude adopted in these conditions. He also studied the physiological modifications of animals in this condition, highlighting the emotional experience of the animal and developing the so-called thalamic theory [9]. Stress studies were significantly developed by Hans Hugo Bruno Selye (1907–1982), an endocrinologist of Hungarian origin, responsible for the fundamental division into distress (negative stress) and eustress (positive stress) based on the pathophysiological responses of living organisms and the degree of intensity of the applied stressor [9,10]. Although these studies have had a significant impact on what will be the foundations of the PNEI, it is necessary to remember that the experimental models of stress used were excessively forced and far from the simple concept of stress: Selye demonstrated the hypertrophy of the adrenal glands of rats subjected to highly harsh treatments (electric shocks, cold, heat, drowning, etc.) in response to massive activation of the hypothalamic-pituitary-adrenal axis, and demonstrated how various harmful agents (stress factors) induced both adrenal hypertrophy such as thymic atrophy. This pathophysiological change took the name of general adaptation syndrome and included a higher incidence of gastric ulcers (Figure 2).

The official birth of the PNEI has a precise date of 1981. The first edition of Psychoneuroimmunology was published by Robert Ader (Ph.D. in Psychology at Cornell University); Ader is considered the father of this discipline, and being a young researcher, in 1975, he demonstrated that the psyche is capable of influencing the immune system. Ader and his colleague, immunologist Nicholas Cohen, showed that if mice were given an immunosuppressive drug along with apple juice, they would suffer from the drug’s effects as soon as they tasted the liquid, even in the absence of the drug. Researcher’s beliefs at the time viewed the immune system as entirely autonomous [7]. It was not a single study that started this discipline. Despite the convergence of several studies that confirmed the interactions between the brain and the immune system, the reaction of the biomedical community, especially immunologists, also regarding other studies by other researchers, such as David Felton and Hugo Besedovsky, on the neuroendocrine regulation of the immune system, was quite adverse. However, the immune system was thought to work autonomously, and there was no way to explain the mechanics of the brain’s influence on it [7].

## 3. Scopes and Limitations of Psychoneuroendocrinoimmunology

The PNEI has been proposed as a bridge in the generation of knowledge between biology, physics and psychology, areas that have been separated until now in the world of science [1,3].

The PNEI is presented as the basis of a radical initiative in the medical-scientific field that, based on a new way of conceiving Biology and Medicine in a holistic knowledge of everything, taking it to its original unit, just as it was at the beginning of human civilization, putting an end in the first place to the separation between Psychology and Medicine and progressively also to that between Philosophy and Natural Sciences, demonstrating as false that typology of Philosophy that pretended to give meaning to reality even before knowing its structure, thus inverting the relationship between Philosophy and Science, not only annulling the subjection of Science to Philosophy but even making Philosophy derive from science, highlighting some anthropological-psychobiological realities, such that they require a general rethinking in the philosophical and theological field [1,11]:(1)The identity between phytochemistry and the spiritual expansion of consciousness, which the culture of the West had opposed each other [12].(2)The distinction between the psyche and Spirit, thus laying the foundations for the elaboration of a Clinic of the Spirit, thus reappropriating as science the dimension of the Spirit, which had been hostage to Religion for centuries and centuries, having reached the self-conscious awareness of being a being of a spiritual nature finally.(3)The reinterpretation of the human being in terms of the unity of a trinity, composed of the biological body, the psyche, and the Spirit [13].

The human body that is taught in all universities is a biological body reduced to a mechanism after selectively amputating all those anatomical-functional structures that make it possible to express pleasure and spiritual self-awareness, that is:(1)Pineal Gland-Melatonin [14](2)Cerebral Cannabinergic System [15](3)Cognitive Activity of the Cerebellum [16](4)Thymus Gland [17](5)Psychoneuroendocrine activity of the heart and coccygeal gland, of a chemoreceptor nature [18].

This whole matter must be rewritten and reinterpreted in a new way, not only the treatment but also the pathophysiology of the major human diseases still incurable. For example, autoimmune diseases must be considered as consequences of an altered psychoneuroendocrine modulation, an expression of the patient’s psychological and spiritual experience and the immune response [19,20].

It is possible to identify at the brain level the existence of two fundamental neurochemical systems, opposite and complementary to each other:The opiate system is connected to the unconscious life, to the pituitary gland or hypophysis located in the brain’s center. This system is activated under stress, pain, anxiety, and irritated ability and induces immunosuppression or disease. It is mediated by catecholamines, adrenal steroids, opioids, endorphins, and dynorphins [21].The cannabinoid system, which is connected to conscious and supraconscious life, allows man to perceive the Universe; it is in relationship with the pineal gland [22] (Figure 3).

It is demonstrable that cannabinoids or substances similar to marijuana, chemical mediators of psychedelic states of dilation of consciousness, promote a form of immunity and therefore are anti-tumors by direct inhibitory action on the proliferation of malignant cells. The man’s identity is inseparable and, at the same time, the different whole between the body, understood as chemistry, and the Spirit: everything chemical affects the affective-spiritual life, and every psychospiritual event induces neuroimmunochemical effects. Therefore, we can affirm that the state of pleasure immunostimulants, thus enhancing the body’s natural resistance to disease, the role of the immune system is not to defend us from the outside but to maintain one’s identity [22].

Therefore, it is not the production of endorphins that makes the perception of pleasure possible and improves immunity, but the production of cannabinoids and pineal indoles such as melatonin [23], showing a broad relationship between the pineal gland and the PNEI. This is similar to the double regulation in the human body: the first, according to the laws of metabolic-biological nature, is exerted by the hypophysis or pituitary gland, under nature and according to its rules; the second about the universal laws exerted by the pineal gland. The pineal gland, already indicated in the past as the seat of the soul by Galen and Descartes, represents the point of union between these two types of regulation [10,11,22].

The heart of people who do not feel pleasure secretes diendothelin-1 (ET 1), which has inhibitory activity on both the pineal and coccygeal glands, and activates the sympathetic system, thus predisposing them to hypertension and myocardial ischemia inducing immunosuppression. In contrast, in a person capable of feeling pleasure, the secretion of atrial natriuretic hormone (ANP) is generated, which stimulates both the pineal gland and the coccygeal gland and activates the parasympathetic system. The action of the sympathetic neuro-vegetative system induces immunostimulation and therefore has an anti-tumor effect [10,11,22] (Figure 4).

The PNEI has developed a valid theoretical model capable of leading to a new diagnosis and a new way of conceiving therapies for human pathologies since it is necessary to associate science with philosophy [8,11]. Even the so-called psychosomatic medicine [24], which has kept away from the rapid evolution of biological knowledge in recent years, has contributed little to understanding the bio-psycho-spiritual singularity of the human being, limiting itself mainly to the evaluation of the nervous system and neuro-vegetative effects. Instead, it almost wholly neglects the knowledge derived from modern psychoneuroendocrinology and psychoneuroendocrineimmunology [11,24,25]. Therefore, with the PNEI, a model of research and interpretation of health and disease emerges that sees the human body as a structured and interconnected unit, where the psychic and biological systems mutually condition each other [25].

The PNEI shows that everything is connected with everything; everything is unity in the biology of the human body, with the various molecules having a plurality of actions, both metabolic and emotional. The PNEI thus forms a new and supreme medical specialization, which in turn presupposes three specialties: (a) immunology; (b) endocrinology; (c) and neurology [24].

Over the years, however, almost all the research carried out has not been able to go beyond the evidence of the reciprocal effects between cytokines and neurons, the point of getting lost in a true labyrinth where everything seems to stimulate everything else, canceling out thus the possibility of developing a science capable of renewing the diagnosis and treatment of human diseases. This is due to the lack of an effective scientific method and adequate human-philosophical preparation, without which it is impossible to make the PNEI clinically relevant [24,26].

## 4. Role of Melatonin in the Neuroendocrine and Immune Systems

Melatonin (MLT) is an ancient molecule traced back to the origin of life. One of the main functions of MLT is to function as a free radical scavenger cone; other functions include the regulation of sleep cycles, the modulation of circadian cycles, enhancement of immunity, and as an oncostatic agent, regulating circadian rhythm, immune and metabolism regulation, antioxidant, anti-aging, and anti-tumor effects [27], prevent cell death, reduce inflammation [28], block calcium channels [29,30], restore autophagy [31] and mitophagy [32], among others [33]. MLT is an indolamine produced in the pineal gland, generated and released in a circadian manner [34]. In humans, the level of MLT increases at dusk, with a higher peak between 2 and 4 am, descending in the second part of the night [34]. In humans and animals, the embryo and fetus depend on maternal MLT, as g. pineal matures and fully develops after birth. MLT crosses all physiological barriers, including the placental barrier, without experiencing any structural or functional modification, and this neurohormone is involved in the placental function itself [34,35]. In humans, the hypothalamic suprachiasmatic nucleus (SCN) expresses receptors to MLT in the fetus and adults. Maternal MLT enters the transplacental fetal circulation by providing photoperiodic information to the fetus [34,35]. MLT concentrations are increased in the maternal circulation during pregnancy, with a peak at the end of pregnancy [35]. The development of fetal sleep patterns in late pregnancy is due to MLT functioning as a regulatory factor, a typical sleep pattern intimately related to neurodevelopment, and providing strong evidence that MLT is involved in fetal neuroprotection [34].

In lower vertebrates, MLT secretion begins early, during embryonic development. The human fetus or newborn does not produce its MLT; it depends on the hormone supplied by the mother via the placenta or milk. Circadian functions in full-term newborn children (MLT secretion, sleep-wake rhythms, body temperature rhythms) do not exhibit circadian variation until 9-12 weeks of postnatal life [34]. At least two types of membrane receptors have been described as MLT, Mel1, and Mel2. Of the MT1 type, there are three subtypes Mel1a, Mel1b, and Mel1c. Mel1a and Mel1b are referred to as MT1 and MT2. The Mel1c receptor subtype is not expressed in mammals [34]. MLT receptors have been identified in embryos and fetuses in the nervous system and peripheral organs, especially in cells of the endocrine system [34]. MLT has been associated with the reproduction process in vertebrates, particularly in seasonal breeding [36].

MLT has been associated with the reproduction process in vertebrates, particularly in seasonal breeding [36]. MLT can regulate several of the reproductive cycles in humans; the pulsatile release of GnRH and the increase in the pulse rate of gonadotrophins are observed to be higher at night during puberty [36], and the elevation in the monthly secretion of LH and FSH by ovulation occurs mainly during the last hours of the dark phase [36,37]. MT1 receptor mRNA has been found in pituitary *pars tuberalis* in rats, as well as the expression of this same receptor in the *pars distalis* in the gonadotrophic fraction [37]. In the anterior pituitary gland, the MLT mediates the effects of the photoperiod, acting mainly on two types of secretory cells, lactotrophs, which secrete prolactin, and gonadotrophs, which secrete two gonadotrophins, luteinizing hormone (LH) and follicle-stimulating hormone (FSH) [37].

The neuroendocrine and immune systems are closely interrelated; products secreted by the neuroendocrine system can affect the immune system and vice versa [38]. At the level of the immune system, MLT shows contradictory effects, and it can exert both inflammatory and anti-inflammatory effects. Proinflammatory effects related to increased resistance of the organism to pathogens have been reported, as well as anti-inflammatory effects in cases of sepsis, cerebral ischemia-reperfusion, and certain neurodegenerative diseases [38]. MLT has been shown to have five family receptors in mammals, two of them are membrane receptors (MT1 and MT2), and the remaining three are nuclear receptors, ROR-α, ROR-β, and ROR-γ [38,39]. The MT1 receptor is expressed in lymphoid tissue and actively participates in the regulation of the inflammatory response. In contrast, the nuclear receptors ROR-α and ROR-γ play a predominant role in the expression of cytokines such as IL-17A, IL-17F, and IL-23R, as well as the chemokines CCL20 and CCR6 [38]. MLT receptors and their synthases have been reported to be expressed by macrophages. They regulate the cellular differentiation pathways of these cells, which is directly related to the immunoregulatory effect of MLT on macrophages in diseases such as cancer and rheumatoid arthritis [38]. So, MLT is a ubiquitous hormone with pleiotropic effects on the function of immune cells (Table 1).

The pineal gland is directly involved in innate and adaptive immune responses, the expression of MHC-II in antigen-presenting cells or APCs (macrophages and dendritic cells), as well as in peritoneal macrophages is increased after MLT supplementation; it has been suggested that this neurohormone can prevent age-related changes in the immune system [38,39]. MLT has been reported to play an anti-inflammatory role in innate immune activation, inhibit the de-acetylation of NF-ĸ B by sirtuin1, inhibit NF-ĸ B activity in the septic mouse, reduce the proinflammatory response mediated by this same transcription factor, and restore mitochondrial homeostasis [38]. MLT can inhibit the p38 pathway, an inflammatory pathway of MAPKs in breast cancer cells while having anti-invasion and anti-metastasis properties [39] (Figure 5).

In humans, the pineal gland’s functions overlap with the functions of melatonin [109]. MLT has been considered neuroprotective against various neurodegenerative and immune diseases due to its wide range of biological functions. Major depressive disorder is a common, chronic, and severe life-threatening disorder that could frequently present in neurodegenerative diseases such as Parkinson’s or Alzheimer’s [31,110,111]. Pathophysiology is characterized by increased oxidative stress and increased levels of proinflammatory cytokines, including tumor necrosis factor (TNF)-α, interleukin (IL)-6, and IL-1β, stimulating astrocytes and microglial cells, leading to the development of depressive symptoms [31]. The MLT could act as an antidepressant via MLT receptor MT2 and regulate FOX03A signaling on astrocytes and microglial cells [31], as well as ameliorated oxidative stress [112,113], neuroinflammation [114], and autophagy deficits [31].

MLT is known to have protective effects on cognitive impairment in neurodegenerative diseases such as Alzheimer’s. Due to restored mitophagy by improving mitophagosome-lysosome fusion via Mcoln1 and ameliorated mitochondrial functions, attenuated amyloid β pathology [32,115,116]. Additionally, it regulates the mTOR expression in the hippocampus while affecting the downstream proinflammatory cytokines [117]. MLT ameliorated oxidative stress-mediated c-Jun N-terminal kinase (JNK) activation, enhanced Akt/extracellular signal-regulated kinase (ERK)/cAMP response element binding protein (CREB) signaling, decreased the levels of apoptotic markers, and increased neuronal survival, promoted cell survival, proliferation, and memory processes [118].

It has been described as a pivotal role for the IL-17 cytokine family in human inflammatory or autoimmune diseases and neurodegenerative diseases [119]. Stimulation of Th17 cells with IL-1β and IL-23 induces local tissue inflammation, mainly mediated by type-17 signature cytokines such as IL-17, IL-22, and granulocyte-macrophage colony-stimulating factor (GM-CSF) [120]. Th17 cells promote neuroinflammation and activation of microglia and astrocytes, actions that contribute to neuronal damage [121] and the development of neuronal diseases. MLT regulates the differentiation of T cells producing IL-17 and induces the synthesis of IL-17A by intact T cells but has little effect on activated cells. Additionally, the high concentration decreased the intracellular expression of IL-17A. So, MLT had a dose-dependent effect [122,123]. Thus, the result of the regulation of all of these molecular mechanisms by MLT is improving cognitive function and ameliorating the pathophysiology of neuronal diseases (Figure 6).

## 5. Conclusions

The pleiotropic characteristics of melatonin lead us to several organic systems with different activities, from an endocrine activity to that of a modulator of nuclear signals through redox activities, scavenger, and what this article is about: how is this indole involved in the development of multiple systems intertwined forming a unit?

Although we have separated and conceptualized each system (Psychological, Nervous, Endocrine, and Immune) as autonomous for decades, the reality is that it integrates the beautiful and complex symphony of life like an orchestra. Worth this last reflection.

Psycho-Neuro-Endocrine-Immunology is a concept that was formed in the 1980s. The objective was to show the links between psychology, neurology, endocrinology, and immunology, integrate all of them, and affirm that psychology is integral to health processes. This is in contrast to psychoendocrinology, which only talks about the psychological and endocrine binomial.

Emotions and psychic phenomena were thought to influence bodily functions for decades, but no one knew how this could be. Thanks to the discovery of neurohormones in the brain, the Cartesian dogma is broken, and the mind–body connection is established; it is now a global and integrated entity. It is necessary to be interested in the health process to understand this field of research. Most diseases are multifactorial, and their appearance depends on the environment, the experience of the individual, and also on the predispositions conferred by their genetic inheritance.

The maintenance of the internal balance of the organism is guaranteed by three primary communication and integration systems: the central and autonomic nervous, endocrine and immune systems. These have long been considered independent, but neuroscience leads us to question this claim and shows that they communicate with each other very precisely. Therefore, it is necessary to consider the relationship between the psyche, the neuroendocrine system, and the immune system as a unit in an integrated network. When it works harmoniously, it ensures psychosomatic well-being. An imbalance of this integrated network will lead to the appearance of psychosomatic illnesses.

The communication between systems is orchestrated by chemical messengers that transmit information, and in these, melatonin plays an important role. A dialogue is established between cells, organs, and tissues in a common biological language. The blood and the nerves transport the information to maintain the optimal functioning of the organism. The nervous system uses a transmission of electrical signals and neurotransmitters. The endocrine system uses signal molecules, the hormones that circulate and transmit specific signals at a distance to their target organ. The immune system sends messages through cells that spread throughout the body and locally produce active molecules of cytokines and antibodies, melatonin pleiotropically plays most of the roles mentioned, and thus these three units interact closely, and it has been possible to highlight the relationships between psychic life and immune defenses, the influence of psyche on the endocrine system, and that immune cells have all the conditions that allow them to be susceptible to the effect of the brain and hormones. It is a dynamic concept that allows for multiple and variable modes of communication between systems. This is why stress and emotional states significantly affect immune function and can cause physical and psychological changes.

## Figures and Tables

**Figure 1 molecules-27-04888-f001:**
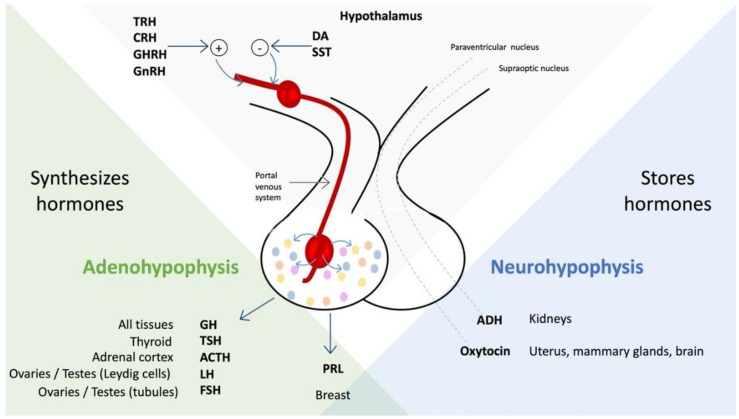
Relationships among hypothalamic hormones, pituitary hormones, and target organs. Neurohypophysis secretion is controlled by nerve signals originating from the hypothalamus, unlike adenohypophysis secretion, which is controlled by hormones or releasing factors from the median eminence. Hormones released from the adenohypophysis are transported to their target organs through the bloodstream. Abbreviations: TRH—thyrotropin-releasing hormone; CRH—corticotropin-releasing hormone; GHRH—growth hormone-releasing hormone; GnRH—gonadotropin-releasing hormone; DA—dopamine; SST—somatostatin; GH—growth hormone; TSH—thyroid-stimulating hormone; ACTH—adrenocorticotropic hormone; LH—luteinizing hormone; FSH—follicle-stimulating hormone; ADH—antidiuretic hormone.

**Figure 2 molecules-27-04888-f002:**
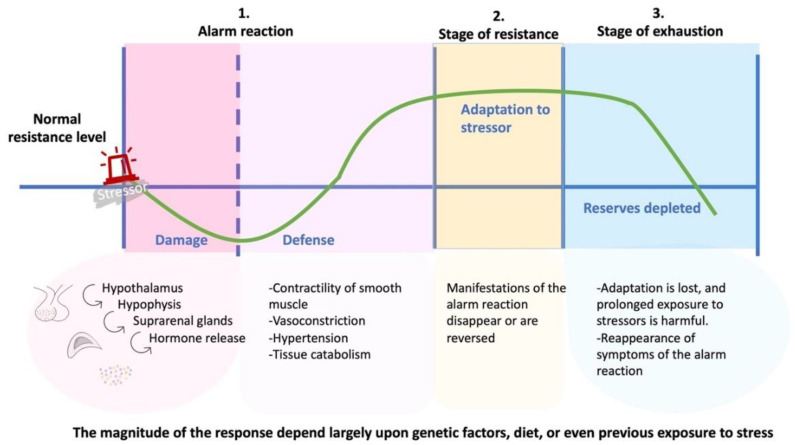
General adaptation syndrome. The general adaptation syndrome is divided into three phases. In the presence of a stressor, phase 1 begins, where the body activates the hypothalamic-pituitary-adrenal axis with the consequent release of hormones to defend itself; in phase 2, symptoms are reduced, and anabolic functions are increased. Phase 3 is exhaustion.

**Figure 3 molecules-27-04888-f003:**
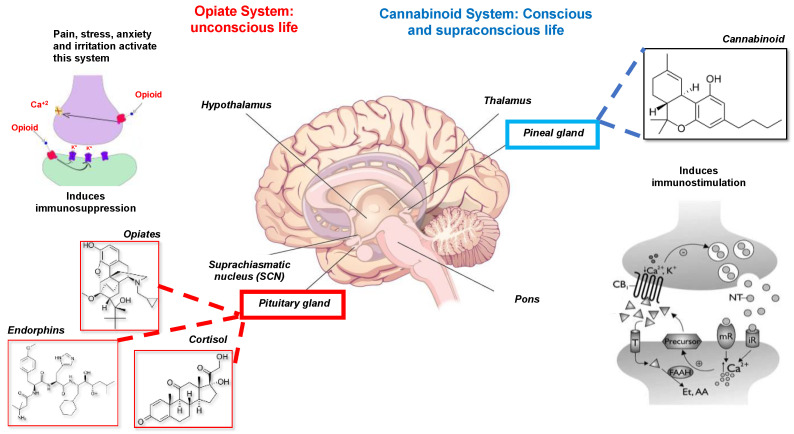
Opiates and cannabinoid system. The activation of the opiate system results in immunosuppression; in contrast, the cannabinoid system results in immunostimulation. Torres-Sanchez ED modified this image, which was obtained from Creative Commons BY based on an unknown author.

**Figure 4 molecules-27-04888-f004:**
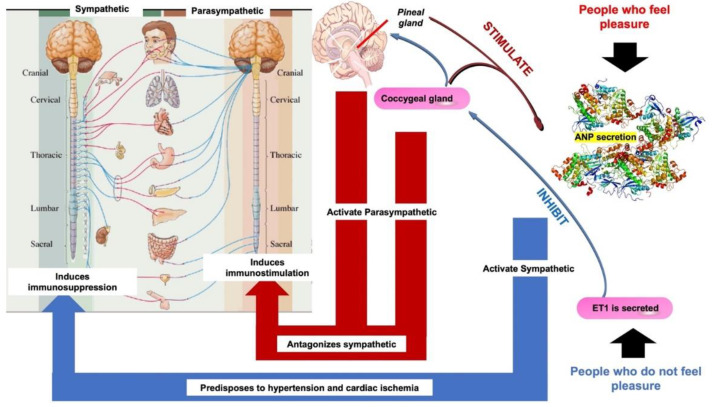
The effect of pleasure on the sympathetic and parasympathetic response induces immunosuppression or immunostimulation, respectively. ED Torres-Sánchez modified this image, obtained from the Creative Commons BY base of an unknown author.

**Figure 5 molecules-27-04888-f005:**
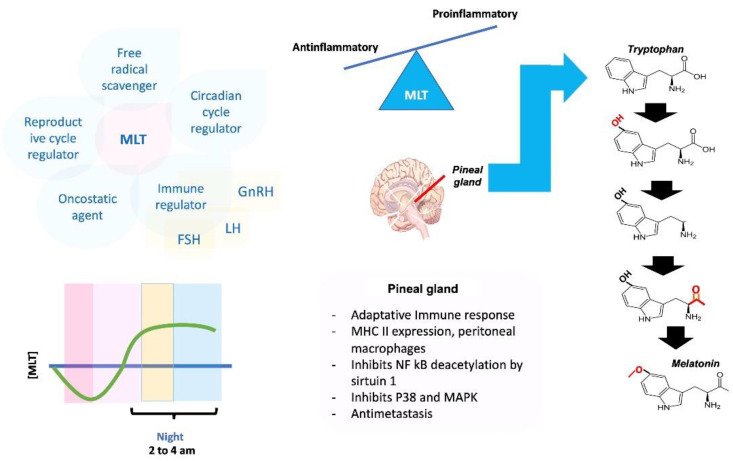
MLT is synthesized in the pineal gland. It has proinflammatory and anti-inflammatory effects and multiple functions, among which the adaptive immune response stands out. ED Torres-Sánchez modified this image, obtained from the Creative Commons BY base of an unknown author.

**Figure 6 molecules-27-04888-f006:**
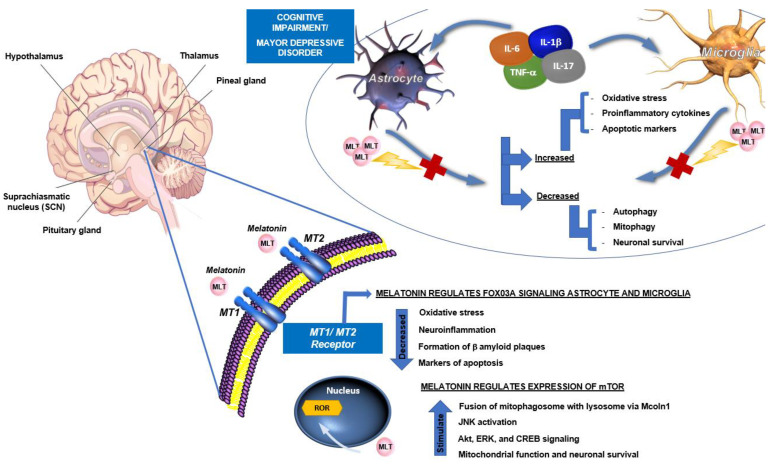
Effects of melatonin in major depressive disorder and cognitive impairments. The pathological process of neurological diseases involves an increase in oxidative stress and an increase in proinflammatory cytokines that stimulate astrocytes and activate microglia. In contrast, by interacting with M2 receptors, melatonin decreases oxidative stress and neuroinflammation. ED Torres-Sánchez modified this image, obtained from the Creative Commons BY base of an unknown author and QIAGEN’s original.

**Table 1 molecules-27-04888-t001:** Abbreviations: MT1/MT2—receptor of melatonin 1/receptor of melatonin 2; ROR—retinoic acid receptor-related orphan receptor; COX-2—cyclooxygenase 2; iNOS—inducible nitric oxide synthase; NF-κB—nuclear factor kappa-light-chain-enhancer of activated B cells; STATs—signal transducer and activator of transcription; NLRP3—NLR family pyrin domain containing 3; α-KG—ketoglutarate alpha; ROS—reactive oxygen species; miR-155—micro RNA 155; miR34a—micro RNA 34a; miR23a—micro RNA 23a; miR27a—micro RNA 27a; miR24-2—micro RNA 24-2; IL-1β—interleukin 1 beta; IL-6—interleukin 6; CXCL8—C-X-C motif chemokine ligand 8; CCL2—C-C motif chemokine ligand 2; CCL4—C-C motif chemokine ligand 4; MMP9—matrix metalloproteinase 9; MAPK—mitogen-activated protein kinase; Akt—protein kinase B; TNF-α—tumor necrosis factor alpha; IL-8—interleukin 8; ERK—extracellular signal-regulated kinase; IL-2—interleukin 2; IL-4—interleukin 4; INF-γ—interferon gamma; CD62L—cluster of differentiation 62 ligand; CD8+—cluster of differentiation 8 positive on T cell; AC—adenylyl cyclase; PKA—protein kinase A; PLC—phospholipase C; PKC—protein kinase C; MHC-II—major histocompatibility complex class II; TLR3—toll receptor 3; p38 MAP—p38 mitogen-activated protein kinases; JNK—c-Jun N-terminal kinase; CD95—cluster of differentiation 95; IL-12—interleukin 12.

Immune Cell	Effects of Melatonin	References
Innate System		
Monocyte	Activates cells through PKC	[40]
	Increases the number of cells	[41]
	Induces ROS production	[40]
	Induces the cytokine production: IL-1, IL-6, TNF-α	[42]
	Attenuates binding activity	[43]
Macrophage	Influences the phenotype polarization	[44,45,46]
	Preventes COX-2 and iNOS activation	[47]
	Regulates signaling pathways: NF-κB, STATs, NLRP3/ caspase-1	[48,49,50]
	Regulates cellular metabolic pathways: α-KG, ROS	[51,52,53]
	Mitochondrial dynamics and mitophagy	[54]
	Regulates the expression of miRNAs: miR-155, miR34a, miR23a, miR27a, miR24-2	[55,56]
	Expression of cytokines: IL-1b, IL-6	[57]
	Could synthesize melatonin	[58]
Dendritic cell	Increases the secretory activity through raised endosomal compartments	[59]
	Increases in the number of cells and diameter	[59,60]
	Could synthesize melatonin	[58,61]
Neutrophil	Protects neutrophils from oxidative stress-induced apoptosis	[62]
	Reduces ROS generation	[62,63]
	Restores their functions: phagocytosis, degranulation, NETosis	[62,64]
	Suppresses the release of CXCL8, CCL2, CCL4, and MMP9 by blockage of p38 MAPK and Akt signaling	[65]
	Suppresses the release of cytokines: TNF-a, IL-8	[66]
	Modulates the migration through the endothelial layer by blocking the ERK phosphorylation signal	[67,68]
	Reduces the microbicidal activity	[69]
Eosinophil	Suppresses eosinophils activity	[70]
	Decreases eosinophil number	[71,72]
Mast cell	Avoid degranulation	[73]
	Regulates differentiation and cellular proliferation	[74,75,76,77]
	Restores circadian rhythms	[78]
	Reduces the cytotoxicity	[79]
	Inhibits the expression of proinflammatory cytokines and COX2	[75,80]
	Could synthesize melatonin	[81]
Adaptative system		
T cell	Decrease migration	[82,83]
	Release cytokines: IL-2, IL-4, IL-6, INF-g	[84,85,86]
	Promotes the balance in subpopulations T cell-mediated immune responses	[87,88,89,90,91]
	Regulates the levels of surface CD62L on CD8+ cells	[92]
	Regulates cell proliferation by AC/ PKA and PLC/ PKC pathways	[41,93,94,95,96,97]
	Enhances antigen presentation by macrophages to T cells by increasing the expression of MHC-II	[98]
	Inhibits the expression of TLR3, p38 MAP, JNK, and MAPK/ NF-κ B pathways	[38]
	Regulates the expression of CD95 ligand	[99]
B cell	Induces the cellular activation	[38]
	Promotes the cell proliferation	[100,101,102]
	Promotes the antibody production	[103,104]
	Represses 5-lipoxygenase gene	[105]
Natural killer	Increases the number of cells	[93]
	Promotes the expression of cytokines: IL-2, IL-12, IL-6, INF-g	[94,106,107]
	Modulates cell activity	[108]
	Regulates the levels of surface CD62L	[92]

## Data Availability

Not applicable.

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
