# Peer review of "Psycho-Neuro-Endocrine-Immunology: A Role for Melatonin in This New Paradigm"

_molecules, 2022, doi:10.3390/molecules27154888_

Round 1

Reviewer 1 Report

1. p.1, introduction: as ‘’immuno’’ represents relation to immune system, the participation of immune system should be explained in the first paragraph.

2. As hypothalamus is usually referred to ‘’Master gland’’, it would be better to highlight its importance and relationship to psychoneuroendocrinoimmunology at the beginning of second paragraph, introduction.

3.what are the major difference between psychoendocrinology and psychoneuroendocrinoimmunology?

4. The discussion of hypothalamus and other neuroanatomical systems like opiate and cannabinoid systems should be grouped together.

5. Part 3, scopes and limitations of psychoneuroendocrinoimunology: the first paragraph mentions that the specialty may ‘’putting an end to …separation between psychology and medicine …’’ may be too farfetched in the context of science.

6. P. 5, line 190: ‘’The PNEI demonstrates….’’ While the previous paragraph is discussing Opiated and cannabinoid system, it is not suitable to switch the subject to PNEI suddenly. This would mix up the discussion of biological system function with the specialty.

Author Response

Thanks for the comments; they have helped us correct and strengthen this manuscript.

All the suggestions were accepted and corrected, in the same way, a paragraph is added in the conclusions section indicating the difference between psychoendocrinology and psychoneuroendocrinoimmunology

Reviewer 2 Report

I have some observations. The first regards the TITLE of the paper. It should be better reformulated as "Psycho-Neuro-Endocrine-Immunology: a role for melatonin in this new paradigm" since there are manuals on the PNEI and a title for a review article is better formulated accordingly. The specific additions of the paper are indeed on Melatonin.

In the references should be listed at least the recent Textbook by Bottaccioli F. and Bottaccioli A.G.

Psychoneuroendocrinoimmunology and the science of integrated medical treatment. The manual

Publisher EDRA 2020  EAN 9788821449222

This manual is available in italian, in spanish and in english editions and is currently the most comprehensive and updated manual in the field and should be widely known by all the scientists in the field. A reference to to this Textbook is mandatory, in my opinion.

Minor remarks:

line 129: Hugo Besedovsls (mis-spelled) his name is Hugo Besedowsky

line 163: This must be rewritten... I would change into "This whole matter must be rewritten"

lines 167-169: concept needs a better explanation (non understandable as it is)

line 182: Opiated and cannabinoid system.. should be instead "Opiates and cannabiboid systems"

line 192: "to her rules" should be instead "to its rules"

lines 199-203: concept needs a better explanation (non well understandable as it is)

Author Response

Thanks for the comments; they have helped us correct and strengthen this manuscript.

The suggested title seemed to us very successful, so we changed it. The manual you requested to reference was an excellent compendium of information, for which it was cited several times throughout the article.

All the remarks were accepted and corrected.

Reviewer 3 Report

This is a rather interesting review about new interdisciplinary directions. The search for new mechanisms of modulation of neuroinflammation are relevant for almost all diseases of the CNS. The recent evidence suggests that neuroinflammation may underlie pathogenesis not only demyelinating diseases of the CNS, but also neurodegenerative diseases (Parkinson's disease), psychiatric disease (schizophrenia, depression), cognitive impairments. In this regard, I may recommend describe the role of melatonin in the development of depression and cognitive impairment. I also recommend describe the influence of melatonin on Th17-cells function, which are considered as a one of the crucial  factor of neuroinflammation. In addition, I recommend making a table which summarizes the effects of melatonin on the cells of both innate and adaptive immune system. 

Author Response

Thanks for the comments; they have helped us correct and strengthen this manuscript.

All suggestions were accepted and corrected in the paper, adding the information and the requested table; likewise, an extra image was added to exemplify the melatonin information.

Round 2

Reviewer 3 Report

Authors reworked the manuscript according to all reviewer's suggestions. I may recommend to accept the manuscript in the present form.